# The Role of Electrochemical Skin Conductance as a Screening Test of Cardiovascular Autonomic Neuropathy in Patients with Parkinson’s Disease

**DOI:** 10.3390/ijerph17217751

**Published:** 2020-10-23

**Authors:** Chih-Cheng Huang, Yun-Ru Lai, Chia-Yi Lien, Ben-Chung Cheng, Nai-Wen Tsai, Cheng-Hsien Lu

**Affiliations:** 1Departments of Neurology, Kaohsiung Chang Gung Memorial Hospital, Chang Gung University College of Medicine, Kaohsiung 833, Taiwan; hjc2828@gmail.com (C.-C.H.); yunrulai@cgmh.org.tw (Y.-R.L.); u9301024@cgmh.org.tw (C.-Y.L.); tsainw@yahoo.com.tw (N.-W.T.); 2Department of Biological Science, National Sun Yat-Sen University, Kaohsiung 804, Taiwan; benzmcl@gmail.com; 3Department of Internal Medicine and Center for Shockwave Medicine and Tissue Engineering, Kaohsiung Chang Gung Memorial Hospital, Chang Gung University College of Medicine, Kaohsiung 833, Taiwan; 4Department of Neurology, Xiamen Chang Gung Memorial Hospital, Xiamen 361028, China

**Keywords:** electrochemical skin conductance, cardiovascular autonomic neuropathy, composite autonomic scoring scale, Parkinson’s disease

## Abstract

Autonomic disorders have been recognized as an important non-motor feature in Parkinson’s disease (PD). However, there is a paucity of information on the presence and severity of cardiovascular autonomic neuropathy (CAN) among different motor phenotypes. The aims of this study were to examine the feasibility of electrochemical skin conductance (ESC) measured by Sudoscan as a screening service for CAN in patients with PD and investigate the severity of CAN among different motor phenotypes. Design: This was a cross-sectional observational study that enrolled 63 patients with PD. Patients were divided into three phenotypes, postural instability/gait difficulty (PIGD), tremor-dominant (TD), and akinetic-rigid (AR), according to their motor symptoms. Cardiovascular autonomic function was measured, and the presence and severity of CAN was determined according to the composite autonomic scoring scale (CASS). Functional scores were measured by the Hoehn and Yahr (HY) stage and the Unified Parkinson’s Disease Rating Scale (UPDRS). The median HY stage was 2.0 [1.5, 3.0]. Median UPDRS total score was 23.0 (17.5, 30.5), 10.0 (6.0, 11.0) and 14.0 (6.3, 23.8) in groups of PIGD, TD and AR, respectively (*p* = 0.001). Mean CASS was 1.7 ± 1.3, 0.6 ± 0.4, and 1.8 ± 1.5 in groups of PIGD, TD and AR, respectively (*p* = 0.204). Although the ESC was not strongly associated with the cardiovascular autonomic parameters, the CAN risk score provided by Sudoscan significantly correlated with parameters of cardiovascular autonomic function, including heart rate response to deep breathing (HR_DB), Valsalva ratio (VR), and baroreflex sensitivity (BRS). By receiver-operating characteristic (ROC) analysis, if a patient’s CAN risk score is higher than 33.5 (%), it is recommended to be aware of the presence of CAN even in PD patients who are asymptomatic. The area under ROC curve was 0.704. Based on our results, CAN risk score may be used for screening of CAN in patients with PD before resorting to the more sophisticated and specific, but ultimately more time-consuming, complete autonomic function testing.

## 1. Introduction

Parkinson’s disease (PD) is a progressive neurodegenerative disorder manifested by a broad spectrum of motor and non-motor features [1,2]. According to the motor symptoms, PD is divided into three clinical phenotypes including postural instability/gait difficulty (PIGD), tremor-dominant (TD), and akinetic-rigid (AR) subtypes [3,4]. The TD subtype has a more benign course of disease with a slower rate of progression compared to the PIGD subtype which is associated with faster cognitive decline, a higher prevalence of non-motor symptoms, and a generally faster progression [5,6]. Non-motor symptoms further provide additional prognostic implications beyond TD versus PIGD motor subtypes [7]. To date, there is a paucity of information on the overall severity of cardiovascular autonomic neuropathy (CAN) using the composite autonomic scoring scale (CASS) [8] among different motor phenotypes.

Autonomic disorders have been recognized as an important non-motor feature in PD [9,10]. In contrast to multiple system atrophy (MSA), which is predominantly a preganglionic disorder, autonomic pathology in PD is primarily postganglionic [11]. Therefore, physiological autonomic testing serves as an important tool in differentiating PD from MSA [12]. The test battery developed by Low at Mayo Clinic covers sympathetic sudomotor, cardiovagal, and adrenergic domains of autonomic function [13] and has been widely accepted as a standard procedure in testing autonomic function for patients with PD. However, the test for sympathetic sudomotor function in this test battery, either a quantitative sudomotor axon reflex test or a thermoregulatory sweat test, is not commonly available. Sudoscan (Impeto Medical, Paris, France) is a recently developed device that accurately and quickly assesses sudomotor function using reverse iontophoresis and chronoamperometry [14] through the measurement of electrochemical skin conductance (ESC) of the hands and feet. In addition to ESC, the Sudoscan device also provides a score, which integrates the measured ESC with age and body mass index (BMI), reflecting the risk of CAN for the tested patient.

Several studies have confirmed that ESC can serve as an indicator of sudomotor function. Studies done in patients with diabetes mellitus (DM) showed that ESC could be a good screening test for CAN [15,16]. In other words, the autonomic impairment in the sympathetic sudomotor and cardiovascular domain may be parallel in patients with DM. There are few similar studies done in patients with PD, and the conclusions are controversial [17,18].

We tested the hypothesis that ESC measurement as well as CAN risk score provided by Sudoscan could predict not only the presence of CAN, but also its severity in patients with PD. The successful translation of these approaches to clinics may offer the awareness of associated CAN in patients with PD to improve the long-term outcome for these patients.

## 2. Patients and Methods

### 2.1. Study Design

This single-center hospital-based cross-sectional observational study enrolled 63 patients with PD from Chang Gung Memorial Hospital-Kaohsiung, a tertiary medical center and the main referral hospital serving a population of three million in southern Taiwan. All participants received verbal and written information about the purpose and process of our research which was approved by the hospital’s institutional Review Committees on Human Research (IRB 201601640B0 and 201702037B0), and the informed consent form was signed.

### 2.2. Inclusion and Exclusion Criteria

A definite diagnosis of idiopathic PD was given according to the United Kingdom Parkinson’s Disease Society Brain Bank clinical diagnostic criteria [19] and patients were followed-up at the Neurology Outpatient Clinic for more than six months after titration of their daily anti-Parkinsonian agents to a steady dose in accordance to their clinical symptoms. The exclusion criteria included: (1) newly diagnosed patients with follow-up for less than six months as their anti-Parkinsonian agents were still under adjustment, (2) presence of focal neurological signs not related to PD, (3) active smoker or stopped less than 5 years before, (4) had pulmonary disease, including chronic obstructive pulmonary diseases, bronchial asthma, or active pulmonary disease within one month of the study, (5) suffered from moderate-to-severe heart failure (NYHA class III and IV), (6) had any type of arrhythmia that prevented baroreflex sensitivity (BRS) measurement, or pacemaker implantation due to any cause, or (7) had diabetes mellitus.

### 2.3. Clinical Assessment

All patients underwent complete neurological examinations, and their age at disease onset (or age at the time of the first reported symptom attributable to the disease), sex, body height, body weight, body mass index (BMI), duration of disease (time from onset until follow-up), and years of education were recorded. The daily dose of anti-Parkinsonian agents was converted into a levodopa equivalent dose (LED) [20]. The severity of PD was assessed using the Unified Parkinson’s Disease Rating Scale (UPDRS) and the Hoehn and Yahr stage [21,22]. The UPDRS total score was computed as the sum of UPDRS subcores I, II, and III. Furthermore, the enrolled patients with PD were divided into three clinical phenotypes: PIGD, TD, and AR subtypes according to the UPDRS. The mean PIGD score was the sum of the UPDRS items 13 (falling), 14 (freezing), 15 (walking), 29 (gait), and 30 (postural instability) divided by five, whereas the mean TD score was the sum of the UPDRS items 16 (both arms tremor determined by history), 20 (face and all four limbs tremor at rest), and 21 (both arms action or postural tremor) divided by eight. The average AR score was calculated as the total score of the UPDRS items 22 (rigidity of neck and four extremities), 23 (finger tap), 24 (hand movement), 25 (hand rotation), 26 (feet flexibility), and 31 (body bradykinesia) divided by 14. Accordingly, the PIGD subtype was defined as the patient’s mean PIGD score/mean TD score > 1.0 and mean PIGD score/mean AR score > 1.0. The TD subtype was defined as the patients mean TD score/mean PIGD score ≥ 1.5 and mean TD score/mean AR score ≥ 1.5. Patients who failed to meet the above-mentioned criteria were grouped into the AR subtype [3].

### 2.4. Assessment of Cardiovascular Autonomic Function

All patients with PD underwent standardized evaluation of cardiovascular autonomic function, as described by Low [13]. The deep breathing, Valsalva maneuver (VM) and 5-min head-up tilt (HUT) tests were done. In addition, 5-min resting recording of blood pressure (BP) and heart rate (HR) were done between tests of VM and HUT. These procedures were done between 9:00 am to noon for all patients. No coffee, food, alcohol, or nicotine was permitted 4 h before the tests. Patients on medications known to cause orthostatic hypotension or otherwise affecting the autonomic testing were asked to stop the drug for five half-lives, if it was not harmful to the patient’s well-being.

Through the above autonomic testing, the parameters including HR response to deep breathing (HR_DB), Valsalva ratio (VR), and baroreflex sensitivity measured by VM (BRS_VM) and by sequence method (BRS_seq) were obtained. Then the severity of CAN was graded using CASS [8]. In addition, the parameters of heart rate variability in frequency domain, including components of low frequency (LF) and high frequency (HF), and LF/HF ratio were computed. The detailed methodology for computation of these parameters was described in our previous study [23]. Patients were divided into two groups according to the sum of subscores in cardiovagal and adrenergic domains of CASS. Patients with a sum equal to or larger than two were defined as the CAN group, whereas the remaining patients were grouped as non-CAN [24]. The CASS had a scale from zero to seven points in this study due to lacking sudomotor subscore [25].

### 2.5. Assessment of Sudomotor Function

Assessment of sudomotor function was performed using a Sudoscan device by the previously described method [14]. Briefly, patients placed their hands and feet on stainless steel plates and incremental DC voltage was applied (<4 V). Then ESCs (expressed in μS) were automatically recorded. In addition, an algorithm that integrates ESC with age and BMI has been developed to produce a CAN risk score to estimate the individual’s current CAN risk [16]. The parameter was also provided by Sudoscan. The procedure of Sudoscan was conducted on the same day as the cardiovascular autonomic testing was done. For most patients, ESC was measured immediately after the completion of cardiovascular autonomic testing.

The clinical assessment and autonomic studies were done during the “off” state of medication, which was defined as 8–12 h after the latest dose of anti-Parkinsonian agents.

### 2.6. Statistical Analysis

Data are expressed as mean ± standard deviation (SD) or median (interquartile range (IQR)). Continuous variables between two groups were compared using an independent Student’s t-test for data with Gaussian distribution and the Mann–Whitney U test for data with non-Gaussian distribution. Continuous variables among three groups were compared using ANOVA for data with Gaussian distribution and the Kruskal–Wallis non-parametric test for data with non-Gaussian distribution. The associations between measurements were evaluated by Pearson correlation tests for normally distributed data or by Spearman non-parametric tests for skewed data. Receiver operating characteristic (ROC) curves were generated for mean hand and foot ESC in the presence of CAN, and the areas under the ROC curves was calculated. Statistical significance was set at *p* < 0.05. All statistical analyses were conducted using the IBM SPSS software package, version 17 (IBM, Inc., Armonk, NY).

### 2.7. Ethics Approval

The study was approved by Chang Gung Memorial Hospital’s Institutional Review Committee on Human Research (201601640B0 and 201702037B0).

## 3. Results

### 3.1. Characteristics of the Study Patients

The baseline characteristics of the 63 enrolled patients are listed on Table 1. The disease duration and HY stage were 5.2 ± 3.9 years and 2.0 (1.5, 3.0), respectively. Although age, gender, BMI, and disease duration were similar among the three subgroups, there were significant differences in the clinical scales. The PIGD group had the highest scores followed by the AR group, and the TD group had the lowest ones.

### 3.2. Comparison of Autonomic Function Among Clinical Subgroups

Table 2 shows each parameter of autonomic function. Although the TD group seemed to have better autonomic function, i.e., lower CASS and higher HR_DB, VR, BRS, and ESC, the differences were not significant.

### 3.3. Comparison of Between Groups of CAN and Non-CAN

Table 3 shows the comparison between the groups of CAN and non-CAN in terms of patients’ characteristics, PD stage, UPDRS score, BRS, LF/HF ratio, ESC, and CAN risk score. There were significant differences in HR_DB, VR, BRS_VM, BRS_seq, and CAN risk score between these two groups. As for patients’ characteristics, PD stage, UPDRS score, mean hand and foot ESC, or LF/HF ratio, no significant difference was seen between these two groups.

### 3.4. Correlation Analysis Among ESC, PD Stage, UPDRS, and Autonomic Parameters

Table 4 shows the results of correlation analysis among CASS and patients’ characteristics, functional score, CAN risk score, and ESCs. Except for the significant positive correlation between CASS and age, no significant correlation was found between CASS and disease duration, functional score, CAN risk score, or ESC, respectively.

Table 5 lists the correlation analysis results among ESCs, CAN risk score, PD stage, UPDRS, and cardiovascular autonomic parameters. There was significant negative correlation between hand ESC and UPDRS subscore II and total score (Figure 1A). As for the cardiovascular autonomic parameters, significant correlation only existed between VR and hand ESC (Figure 1B). However, the correlation coefficient indicates a weak positive linear relationship (r < 0.4). In addition, the cardiovagal subscore of CASS had significant negative correlation with hand ESC. CAN risk score significantly correlated with HR_DB, VR, and BRS_seq, respectively (Figure 2A–C). On the contrary, no significant correlation was found between foot ESC and PD stage, UPDRS score, or cardiovascular autonomic parameters.

The area under the ROC curve for mean hand and foot ESCs, and CAN risk score in the presence of CAN were 0.597 (*p* = 0.234, 95% confidence interval (CI) = 0.44–0.753), 0.606 (*p* = 0.194, 95% CI = 0.451–0.760), and 0.704 (*p* = 0.007, 95% CI = 0.572–0.836), respectively. The cut-off value in diagnostic accuracy for CAN risk score in the presence of CAN was 33.5 (sensitivity = 59.3% and specificity =67.6%) (Figure 3)

## 4. Discussion

### Main Findings of Our Study

The American Academy of Neurology summary of evidence-based guidelines for clinicians recommends that the combination of autonomic screening tests in the CASS should be considered to achieve the highest diagnostic accuracy of CAN [26]. To the best of our knowledge, our work is the first study to adopt cardiovagal and adrenergic subscores of the CASS as a measure of CAN, and examine the feasibility of ESC as a screening service of CAN in patients with PD.

Our data showed significantly different clinical scales among the three subgroups of patients with PD. The PIGD group had the worst function, the TD group the best, while the AR group was placed in between. The finding was consistent with the original report suggesting such phenotype classification [3]. Our data did not show a similar difference of autonomic function among these subgroups. Although the TD subgroup had higher values for all autonomic parameters and lower CASS, suggesting better autonomic function than the other two subgroups, the difference was not statistically significant. Further study is warranted to answer the question of whether autonomic function differs among different phenotypes of PD. This study also showed the severity of CAN, demonstrated by CASS, neither significantly correlated with clinical scores nor the ESC values.

There are only a few clinical studies using Sudoscan to evaluate CAN in patients with PD. The study by Xu et al. enrolled 43 patients having later stage PD (mean UPDRS score = 38.5 ± 9.0) with mean age 66.0 ± 8.9 years, and 42 healthy controls [18]. This study showed that ESC of the limbs, especially the hands, was significantly lower in patients with PD than in controls, and was significantly correlated with the results of the scale of autonomic symptoms (SCOPA-AUT) in PD. The report by Popescu et al. enrolled 67 patients with PD (mean UPDRS score = 21.2 ± 11.9) whose mean age was 74.3 years, and 66 aged-match controls [17]. This study showed no significant reduction in ESC in patients with PD compared with controls, and further found weak correlation between foot ESC and modified Hoehn and Yahr Scale. Our study enrolled 63 patients with PD (the median UPDRS score was 18.0) whose mean age was 65.4 ± 9.8 years. According to our results, there was no significant difference of either hand or foot ESC between patients with CAN and without CAN. Our patients were in a relatively early stage, and their cardiovascular impairment was mild in severity. The discrepancy among these three studies may be attributed to the different methodologies (e.g., comparisons between patents with PD and controls, and between PD patients with and without CAN), assessments of CAN (SCOPA-AUT vs. CASS), PD stages of patients (later or earlier stage), and statistical methods.

CAN risk score integrates ESC with age and BMI to estimate the individual’s current CAN risk [16]. Although the algorithm of the risk score was developed in patients with diabetes, it may also be applied to patients with PD according to our data. The CAN risk score had higher correlation coefficients with cardiovascular autonomic function (HR_DB, VR, and BRS_seq) than the original ESC value. On contrary to ESC value, there was significant difference between patients with and without CAN (*p* = 0.029). Our study also showed that CAN risk score could provide good diagnostic accuracy for the presence of CAN in patients with PD [27]. The area under the ROC curve for CAN risk score in predicting the presence of CAN was 0.704.

The results of our study showed no significant correlation between ESC and cardiovascular subscores of CASS. This finding implies that the deterioration of autonomic function in patients with PD may be unparalleled in different domains (sudomotor and cardiovascular). The notion is supported by a previous report by Kim et al. who studied autonomic function, including cardiovagal, adrenergic, and sympathetic sudomotor domains, in patients with PD [9]. They showed that the proportion of abnormal patients with cardiovagal function (HR_DB and VR) differed among the groups divided by each HY stage, whereas adrenergic (HUT) and sympathetic sudomotor function (measured by quantitative sudomotor axon reflex test (QSART)) did not differ significantly among groups. As for the comparison between Sudoscan and other established methods for sudomotor function measurement, Novak has shown that ESC correlated with skin nerve fiber density and sweat gland nerve fiber density [28], but studies thus far have shown little or no correlation between ESC and QSART [29]. Although these tests all evaluate sympathetic sudomotor or small-fiber function, the applied mechanisms and physiologies are totally different for each test.

Previous studies have shown that autonomic pathology in PD is primarily postganglionic [11]. The reduction of ESC was expected to follow a length-dependent pattern. Foot ESC may be more sensitive in showing abnormalities than hand ESC. Nevertheless, it is surprising that our data revealed UPDRS and parameters of cardiovagal function correlated with hand ESC instead of foot ESC although the correlation coefficient indicates a weak positive linear relationship (r < 0.4). The results are consistent with a report by Xu et al. which showed that a decrease in ESC was greater in the hands than in feet for patients with PD [18]. We suggest some possible explanations. First, measurement of ESC may be confounded by local skin factors such as skin temperature, increased thickening of stratum corneum, and the dirt on the skin. These factors are more likely to occur on the feet than on hands thus preventing distinct measurement of ESC. Furthermore, ESC, strictly speaking, is a measurement of sweat gland function, which is assumed to reflect autonomic nerves innervating the gland. However, a study by Duchesne et al. showed weak correlation of ESC with skin biopsy. Therefore, they suggested that mechanisms other than the loss of innervating fibers may be responsible for sweat gland dysfunction in polyneuropathies [30]. Finally, the pathophysiology underlying autonomic neuropathy in PD is not totally understood. A severe non-length-dependent reduction of substance *p*-immunoreactive intraepidermal nerve fibers in patients with PD has been reported by Doppler et al. [31].

There are some limitations in this study. First, although both the clinical assessment and autonomic testing were done in the “off” state, 8–12 h after the latest dose of anti-Parkinsonian agents, lingering effects from the medication on autonomic function, especially sudomotor function cannot be excluded. Second, this was a cross-sectional study, and thus it did not answer the question of whether ESC measured by a Sudoscan is effective as a tool to follow-up sudomotor function for patients with PD. In addition, different ethnic groups may affect the normal values of ESC, which seem to be lower in the Chinese population [32]. A more comprehensive study including normal healthy controls is mandatory. Finally, the patient number in the TD subgroup was limited. Although our results showed that the TD subgroup had better autonomic function and lower CASS, it remained inconclusive whether autonomic function differs among different phenotypes of PD.

## 5. Conclusions

Although the ESC was not strongly associated with the cardiovascular autonomic parameters, CAN risk score significantly correlated with parameters of cardiovascular autonomic function. If a patient’s CAN risk score is higher than 33.5 (%), it is recommended to be aware of the presence of CAN even in PD patients who are asymptomatic. Based on our results, CAN risk score may be used for the screening of CAN in patients with PD before resorting to the more sophisticated and specific, but ultimately more time-consuming, complete autonomic function testing (e.g., CASS).

## Figures and Tables

**Figure 1 ijerph-17-07751-f001:**
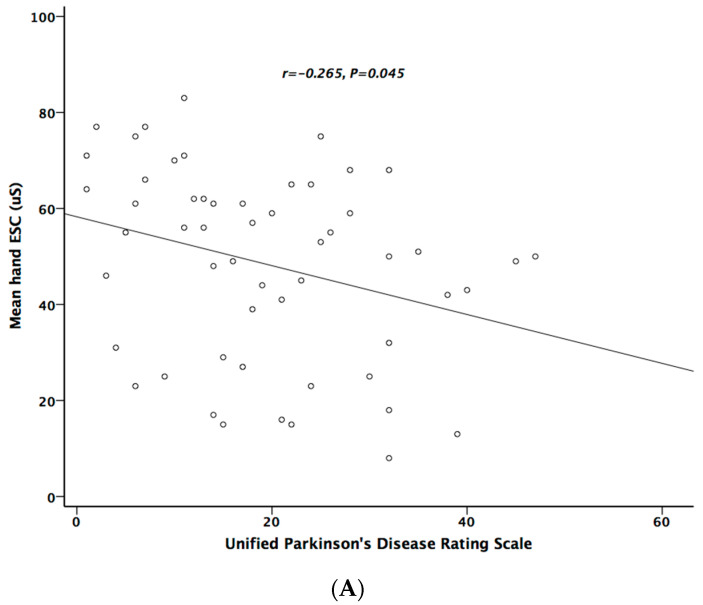
The relationship between electrochemical skin conductance (ESC) of hand and UPDRS (**A**), and between hand ESC and Valsalva ratio (**B**).

**Figure 2 ijerph-17-07751-f002:**
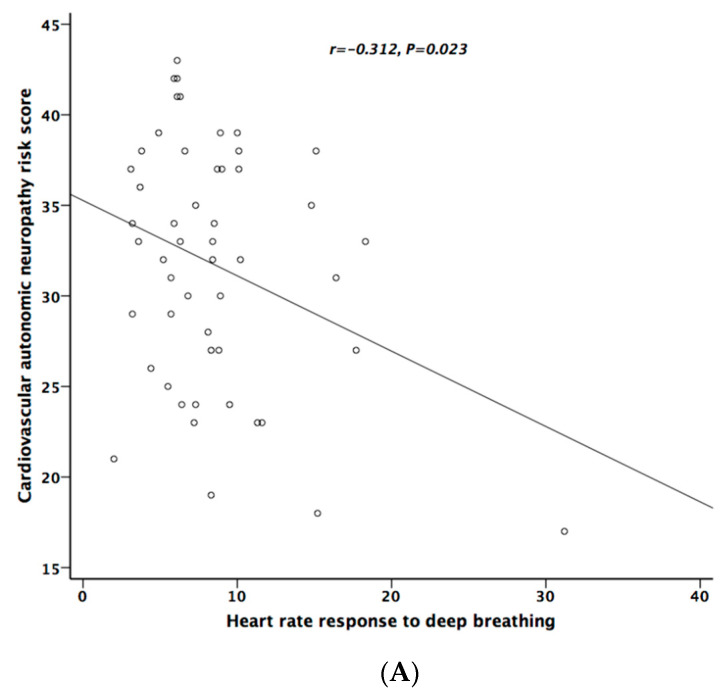
The relationship between cardiovascular autonomic neuropathy (CAN) risk score and heart rate response to deep breathing (**A**), Valsalva ratio (**B**), and baroreflex sensitivity (**C**), respectively.

**Figure 3 ijerph-17-07751-f003:**
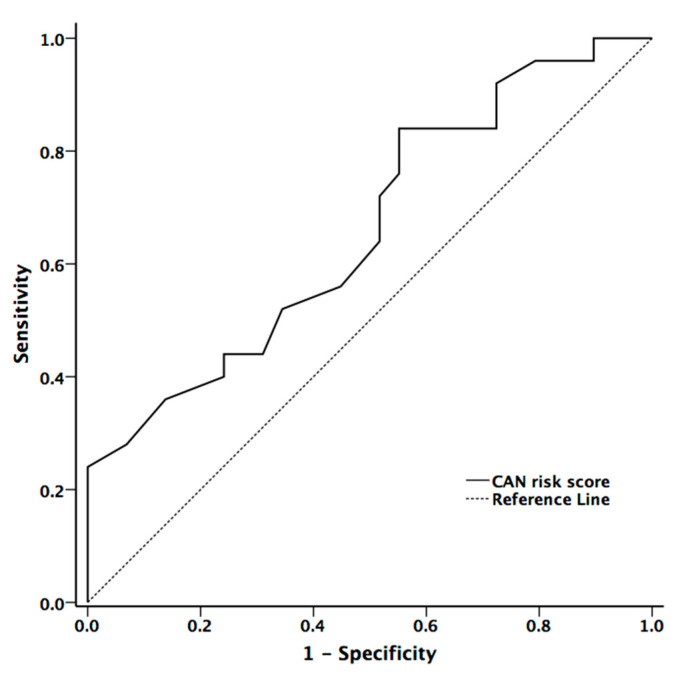
The receiver operator characteristic curve for CAN risk score for diagnostic accuracy for presence of cardiovascular autonomic neuropathy in patients with Parkinson’s disease (PD).

**Table 1 ijerph-17-07751-t001:** Baseline characteristics of Parkinson’s Disease.

	Subtype of Parkinson’s Disease			Total (*n* = 63)	*p*-Value
	PIGD (*n* = 29)	TD (*n* = 7)	AR (*n* = 27)		
Age	65.2 ± 9.1	64.9 ± 10.9	65.7 ± 10.7	65.4 ± 9.8	0.858
Gender (F/M)	17/12	3/4	15/12	35/28	0.75
BMI	25.3 ± 5.3	25.6 ± 3.5	24.6 ± 4.3	24.9 ± 4.7	0.824
Disease duration	6.7 ± 4.7	4.6 ± 2.8	3.7 ± 2.5	5.2 ± 3.9	0.037 *
LED	836 (400, 1210)	425 (260, 663)	578 (455,1050)	613 (375, 1059)	0.085
HY stage	2.5 (2.0, 3.0)	1.5 (1.0, 1.5)	2.0 (1.5, 2.8)	2.0 (1.5, 3.0)	0.002 ***
UPDRS I	1.0 (2.0, 3.0)	0.0 (0.0, 1.0)	1.0 (0.0, 1.8)	1.0 (1.0, 2.0)	0.001 ***
UPDRS II	9.0 (8.0, 10.3)	3.0 (1.0, 6.0)	4.0 (2.3, 8.0)	8.0 (3.8, 10.0)	<0.001 ***
UPDRS III	12.0 (7.8, 17.5)	5.0 (2.0, 7.0)	7.5 (4.3, 8.0)	9.0 (5.0, 16.3)	0.031 *
UPDRS total	23.0 (17.5, 30.5)	10.0 (6.0, 11.0)	14.0 (6.3, 23.8)	18.0 (10.8, 28.0)	0.001 ***

Abbreviations: PIGD, postural instability and gait disturbance; TD, tremor dominant; AR, akinetic rigidity; BMI, body mass index; LED, levodopa equivalent dose; HY, Hoehn and Yahr; UPDRS, Unified Parkinson’s Disease Rating Scale; α = “Total UPDRS” score is the combined sum of parts I, II, and III. β = I. Mentation, behavior, and mood. γ = II. Activities of daily living (ADL). δ = III. Motor examination. **p* < 0.05; ****p* < 0.005.

**Table 2 ijerph-17-07751-t002:** Comparison of cardiovascular autonomic function among subtypes of Parkinson’s disease.

	Subtype of Parkinson’s Disease			Total (*n* = 63)	*p*-Value
	PIGD (*n* = 29)	TD (*n* = 7)	AR (*n* = 27)		
CAN Risk Score	31.4 ± 6.6	30.2 ±7.9	32.8 ± 6.7	31.8 ± 6.7	
Mean hand ESC, µS	47.8 ± 17.0	54.1 ± 23.9	50.1 ± 20.0	48.9 ± 19.3	0.617
Mean leg ESC, µS	58.8 ± 19.0	60.3 ± 25.6	58.5 ± 17.9	58.4 ± 19.2	0.836
CASS	1.7 ± 1.3	0.6 ± 0.4	1.8 ± 1.5	1.7 ± 1.4	0.204
HR_DB	7.1 ± 3.2	13.0 ± 8.7	8.4 ± 4.4	8.3 ± 4.9	0.056
VR	1.35 ± 0.19	1.55 ± 0.26	1.33 ± 0.23	1.36 ± 0.22	0.076
BRS_VM	1.7 ± 0.9	2.2 ± 1.1	2.1 ± 1.1	1.9 ± 1.0	0.262
BRS_seq	8.3 ± 4.9	9.9 ± 3.2	6.7 ± 3.4	7.8 ± 4.2	0.114
LF/HF ratio	0.56 (0.33, 0.95)	1.41 (0.35, 3.03)	0.72 (0.37, 1.68)	0.70 (0.37, 1.40)	0.459

Abbreviations: PIGD, postural instability and gait disturbance; TD, tremor dominant; AR, akinetic rigidity; CAN, cardiovascular autonomic neuropathy; ESC, electrochemical skin conductance; CASS, composite autonomic scoring scale; HR_DB, heart rate response to deep breathing; VR, Valsalva ratio; BRS_VM, baroreflex sensitivity obtained by Valsalva maneuver; BRS_seq, baroreflex sensitivity obtained by sequence method; LF, low frequency; HF, high frequency.

**Table 3 ijerph-17-07751-t003:** Comparison between patients with or without cardiovascular autonomic neuropathy.

	Non-CAN (*N* = 34)	CAN (*N* = 29)	*p*-Value
Age (years)	62.6 ± 10.3	68.7 ± 8.1	0.012 *
Gender (F/M)	18/16	17/12	0.654
Disease duration (years)	5.6 ± 4.3	4.7 ± 3.3	0.378
LED	638 (413, 1277)	613 (338, 1050)	0.517
HY stage	2.0 (1.5, 3.0)	2.0 (1.9, 2.6)	0.693
UPDRS I	1.0 (0.3, 2.0)	1.0 (1.0, 2.0)	0.948
UPDRS II	7.5 (2.0, 10.0)	8.0 (5.0, 9.3)	0.451
UPDRS III	11.0 (4.3, 16.8)	8.0 (5.8, 15.5)	0.975
UPDRS total	18.5 (6.3, 28.0)	17.5 (11.8, 26.8)	0.673
CASS	1 (0,1)	3 (2, 3.5)	<0.001 ***
HR_DB	10.4 ± 4.9	5.6 ± 3.2	<0.001 ***
VR	1.44 ± 0.23	1.29 ± 0.20	0.013 *
BRS_VM	2.3 ± 1.1	1.4 ± 0.6	0.001 ***
BRS_seq	8.9 ± 4.7	6.5 ± 3.0	0.026 *
LF/HF ratio	0.70 (0.42, 1.44)	0.71 (0.31, 1.43)	0.613
Mean hand ESC, µS	50.6 ± 18.6	46.9 ± 20.1	0.445
Mean foot ESC, µS	61.9 ± 18.4	54.3 ± 19.5	0.117
CAN Risk score	30.0 ± 6.6	33.9 ± 6.2	0.029 *

Abbreviations: CAN, cardiovascular autonomic neuropathy; LED, levodopa equivalent dose; HY, Hoehn and Yahr; UPDRS, Unified Parkinson’s Disease Rating Scale; α = “Total UPDRS” score is the combined sum of parts I, II, and III. β = I. Mentation, behavior, and mood. γ = II. Activities of daily living (ADL). δ = III. Motor examination; CASS, composite autonomic scoring scale; HR_DB, heart rate response to deep breathing; VR, Valsalva ratio; BRS_VM, baroreflex sensitivity obtained by Valsalva maneuver; BRS_seq, baroreflex sensitivity obtained by sequence method; LF, low frequency; HF, high frequency; ESC, electrochemical skin conductance; CAN, cardiovascular autonomic neuropathy; **p* < 0.05; ****p* < 0.005.

**Table 4 ijerph-17-07751-t004:** Correlation analysis between composite autonomic scoring scale and parameters of functional outcomes and electrochemical skin conductance.

Variables	Composite Autonomic Scoring Scale	
	r	*p*-Value
Age (years)	0.285	0.035 *
Disease duration (years)	−0.076	0.579
LED	0.010	0.940
HY stage	0.061	0.664
UPDRS I	0.184	0.187
UPDRS II	0.166	0.234
UPDRS III	0.111	0.428
UPDRS total	0.162	0.246
Mean hand ESC, µS	−0.168	0.221
Mean foot ESC, µS	−0.235	0.084
CAN Risk score	0.199	0.17

Abbreviations: r, Correlation coefficient; LED, levodopa equivalent dose; HY, Hoehn and Yahr; ESC, electrochemical skin conductance; CAN, cardiovascular autonomic neuropathy; UPDRS, Unified Parkinson’s Disease Rating Scale; α = “Total UPDRS” score is the combined sum of parts I, II, and III. β = I. Mentation, behavior, and mood. γ = II. Activities of daily living (ADL). δ = III. Motor examination; ESC, electrochemical skin conductance; CAN, cardiovascular autonomic neuropathy; * *p* < 0.05.

**Table 5 ijerph-17-07751-t005:** Correlation analysis of electrochemical skin conductance in patients with Parkinson’s disease.

	Mean Hand ESC, µS		Mean Foot ESC, µS		CAN Risk Score	
	r	*p*-Value	r	*p*-Value	r	*p*-Value
HY stage	−0.194	0.144	−0.124	0.356	−0.067	0.641
UPDRS I	−0.207	0.120	−0.110	0.410	−0.009	0.950
UPDRS II	−0.315	0.016 *	−0.193	0.147	0.074	0.608
UPDRS III	−0.204	0.124	−0.112	0.402	−0.083	0.565
UPDRS total	−0.265	0.045 *	−0.143	0.283	−0.031	0.829
HR_DB	0.217	0.099	0.182	0.167	−0.312	0.023 *
VR	0.280	0.044 *	0.252	0.072	−0.368	0.010 *
BRS_VM	0.133	0.358	0.166	0.248	0.134	0.380
BRS_seq	0.128	0.352	0.068	0.622	−0.404	0.004 ***
Cardiovagal subscore	−0.285	0.043 *	−0.202	0.156	0.160	0.262
Adrenergic subscore	0.061	0.662	−0.120	0.387	0.243	0.080
LF/HF ratio	0.108	0.432	0.146	0.288	−0.246	0.088

Abbreviations: ESC, electrochemical skin conductance; CAN, cardiovascular autonomic neuropathy; r, Correlation coefficient; HY, Hoehn and Yahr; UPDRS, Unified Parkinson’s Disease Rating Scale; α = “Total UPDRS” score is the combined sum of parts I, II, and III. β = I. Mentation, behavior, and mood. γ = II. Activities of daily living (ADL). δ = III. Motor examination; HR_DB, heart rate response to deep breathing; VR, Valsalva ratio; BRS_VM, baroreflex sensitivity obtained by Valsalva maneuver; BRS_seq, baroreflex sensitivity obtained by sequence method; LF, low frequency; HF, high frequency; **p* < 0.05; ****p* < 0.005.

## Data Availability

The datasets used and/or analyzed during the current study are available from the corresponding author on reasonable request.

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
