# Peer review of "The Role of Electrochemical Skin Conductance as a Screening Test of Cardiovascular Autonomic Neuropathy in Patients with Parkinson’s Disease"

_ijerph, 2020, doi:10.3390/ijerph17217751_

Round 1
Reviewer 1 Report
The authors conducted a research evaluating electrochemical skin conductance in Parkinson patients. My concerns:
- The abstract should be rewritten. There is no mention of methods, ie what tools did they use? How many participants were included this study? There are strong statements such as in the last sentence. The authors should use specific phrases like 'it seems', 'the results suggest'... The abstract should be written in a comprehensive way.
- The Methods section are incomprehensible, especially about the definition of cases and controls.
- The authors are kindly requested to conform to an EQUATOR Statement according to the type of their study that should be the first thing to be mentioned in the title.
Author Response
- The abstract should be rewritten. There is no mention of methods, ie what tools did they use? How many participants were included this study? There are strong statements such as in the last sentence. The authors should use specific phrases like 'it seems', 'the results suggest'... The abstract should be written in a comprehensive way.
Answer: The abstract was rewritten according to your suggestion. The tools and the number of participants were mentioned. Especially, the conclusions have been written in a tempered statement.
- The Methods section are incomprehensible, especially about the definition of cases and controls.
Answer: A sub-section of “inclusion and exclusion criteria” was added in the Method section defining the participants. There were no controls in this study.
- The authors are kindly requested to conform to an EQUATOR Statement according to the type of their study that should be the first thing to be mentioned in the title.
Answer: The title of the article cannot be modified in the submission system. We have mentioned the type of the study, a cross-sectional observational study, in the first sentence of the Method section.
Reviewer 2 Report
I have the following comments
1) The introduction of an age and gender-matched control group should enhance the quality of the manuscript.
2) Patients received verbal and written information, but I have not seen expressed that the signed informed consent.
3) Was diagnosis with diabetes mellitus an exclussion criteria. If not, these could affect the results.
4) In the discussion, it should be Popescu et al. [17] instead of Popescu et al. (2019), according to the references style for this journal.
Author Response
- The introduction of an age and gender-matched control group should enhance the quality of the manuscript.
Answer: Thank you for your suggestion. We totally agree that adding a control group does enhance the value of the study and the difference between patients with Parkinson’s disease (PD) and normal subjects can be evaluated. However, the main aim of this study was to see if ESCs measured by Sudoscan can differentiate patients with CAN from patients without CAN in general PD population. Therefore, there was no control group in the original study design.
- Patients received verbal and written information, but I have not seen expressed that the signed informed consent.
Answer: Yes, the informed consent form was signed by the participant. We have added the statement in the last sentence in the “study design” sub-section in the Method section
- Was diagnosis with diabetes mellitus an exclusion criteria. If not, these could affect the results.
Answer: We did exclude patients with diabetes in this study. The statement was added in the exclusion criteria in the Method section.
- In the discussion, it should be Popescu et al. [17] instead of Popescu et al. (2019), according to the references style for this journal.
Answer: The reference style was corrected.

Reviewer 3 Report
Major issues
- The stated hypotheses are: "...that ESC measurement via a Sudoscan could predict not only the presence of CAN, but also its severity in patients with PD", however, the manuscript's "Discussion" and "Conclusions" focus on CAN risk scores. The CAN risk score is not mentioned in the Introduction or description of study objectives. Please revise the manuscript so that the objectives/aims and the Discussion/Conclusions sections are more congruent.
- The tremor-dominant group size was very small, which limits the findings. Consider adding effect size estimates. This small sample of TD patients should be discussed as a limitation, and the findings between PD group subtypes tempered accordingly.
- How could identifying CAN improve outcomes and quality of life for people with PD? I think this statement needs to be tempered, especially since ESC was shown not to correlate with CAN in this study.
Minor points to clarify:
-This manuscript is very difficult to read in view of the use of so many acronyms. In addition, recommend that authors revise the manuscript language in general to reduce the use of passive voice, in order to improve the manuscripts's readability. Also recheck for minor grammatical errors.
-Did participants provide informed consent? It says that they received information about the research project but it is not clear that they consented to participate.
-Line 77: should be "definite" rather than "definitive" diagnosis of Parkinson disease
Author Response
Major issues
- The stated hypotheses are: "...that ESC measurement via a Sudoscan could predict not only the presence of CAN, but also its severity in patients with PD", however, the manuscript's "Discussion" and "Conclusions" focus on CAN risk scores. The CAN risk score is not mentioned in the Introduction or description of study objectives. Please revise the manuscript so that the objectives/aims and the Discussion/Conclusions sections are more congruent.
Answer: We added a sentence introducing CAN risk score, “In addition to ESC, the Sudoscan device also provides a score, which integrates the measured ESC with age and BMI, reflecting the risk of CAN for the tested patient.” in the Introduction section. Furthermore, the statement of the test hypothesis has been modified as: “We tested the hypothesis that ESC measurement as well as CAN risk score provided by a Sudoscan could predict not only the presence of CAN, but…”.
- The tremor-dominant group size was very small, which limits the findings. Consider adding effect size estimates. This small sample of TD patients should be discussed as a limitation, and the findings between PD group subtypes tempered accordingly.
Answer: We have added some statements discussing the small sample size of TD group in the last of the limitation paragraph in the Discussion section.
- How could identifying CAN improve outcomes and quality of life for people with PD? I think this statement needs to be tempered, especially since ESC was shown not to correlate with CAN in this study.
Answer: Thank you for your suggestion. We have modified the statements into more tempered form as following: “The successful translation of these approaches to clinics may offer the promise of improving the long-term outcome for patients with PD.”
Minor points to clarify:
-This manuscript is very difficult to read in view of the use of so many acronyms. In addition, recommend that authors revise the manuscript language in general to reduce the use of passive voice, in order to improve the manuscripts' readability. Also recheck for minor grammatical errors.
Answer: Thanks for your suggestion. We have made a list for the acronyms and full forms that were used in the article to improve the readability. The grammar and spelling have been checked again.
-Did participants provide informed consent? It says that they received information about the research project but it is not clear that they consented to participate.
Answer: Yes, the informed consent form was signed by the participant. We have added the statement in the last sentence in the “study design” sub-section in the Method section
-Line 77: should be "definite" rather than "definitive" diagnosis of Parkinson disease
Answer: It has been corrected into “definite”.
Reviewer 4 Report
1. The topic of PD has received much attention and has been extensively studied; therefore it would be beneficial to readers to add few more sentences in the introduction section on the current state of PD research and standard methods of screening. A google search on PD has given the following papers [1] CSF and blood biomarkers for Parkinson's disease L Parnetti, L Gaetani, P Eusebi, S Paciotti… - The Lancet …, 2019 – Elsevier [2] Gait impairments in Parkinson's disease A Mirelman, P Bonato, R Camicioli, TD Ellis… - The Lancet …, 2019 - Elsevier [3] Postural instability in parkinson's disease: A review B Palakurthi, SP Burugupally - Brain sciences, 2019 - mdpi.com And many more…. 2. The error bars on the ESC measurements are above 75% and the p-value in some of the Tables and figures are above the acceptable value of 0.05. Therefore, it is highly recommended to redo some of the experiments for more significance. This will strengthen the study. 3. Some figures are not fully legible and hard to read. Please redo the figures. 4. The authors are suggested to rewrite the conclusion section to clearly quantify the outcomes of the work. For example, include the numerical values such as the p-value, ESC measurements.Author Response
- The topic of PD has received much attention and has been extensively studied; therefore it would be beneficial to readers to add few more sentences in the introduction section on the current state of PD research and standard methods of screening. A google search on PD has given the following papers [1] CSF and blood biomarkers for Parkinson's disease L Parnetti, L Gaetani, P Eusebi, S Paciotti… - The Lancet …, 2019 – Elsevier [2] Gait impairments in Parkinson's disease A Mirelman, P Bonato, R Camicioli, TD Ellis… - The Lancet …, 2019 - Elsevier [3] Postural instability in parkinson's disease: A review B Palakurthi, SP Burugupally - Brain sciences, 2019 - mdpi.com And many more….
Answer: The aim of this study was to test the validity of ESC measured by Sudoscan as a screening tool for CAN in patients with PD. We did not aim to screen PD in general population.
- The error bars on the ESC measurements are above 75% and the p-value in some of the Tables and figures are above the acceptable value of 0.05. Therefore, it is highly recommended to redo some of the experiments for more significance. This will strengthen the study.
Answer: We re-check the statistical results in Tables and Figures in accordance with the comment for authors.
- Some figures are not fully legible and hard to read. Please redo the figures.
Answer: Thanks for your suggestion. All the figures have been re-done. We added Dr Nai-Wen Tsai, as co-author for his help for the revision of the figures.
- The authors are suggested to rewrite the conclusion section to clearly quantify the outcomes of the work. For example, include the numerical values such as the p-value, ESC measurements.
Answer: In the conclusion section, we rewrite the sentences in accordance with the comment for authors. They are as follows:
If a patient's CAN risk score is higher than 33.5 (%), caution is needed with the presence of CAN even in PD patients who are asymptomatic. Based on our results, CAN risk score may be used for screening of CAN in patients with PD before resorting to the more sophisticated and specific, but ultimately more time-consuming complete autonomic function testing (e.g., CASS).

Reviewer 5 Report
In the article “The Role of Electrochemical Skin Conductance as a Screening Test of Cardiovascular Autonomic Neuropathy in Patients with Parkinson’s Disease” by Huang et al., the authors correlated the severity of cardiovascular autonomic neuropathy (CAN) with severity of Parkinson’s Disease (PD) using electrochemical skin conductance (ESC). They concluded that CAN severity did not correlate with the PD clinical scores or ESC parameters.
Following are my comments:
- The abstract did not mention the conclusion and significance of the study.
- With reference to the statement made in the introduction “There are few similar studies done in patients with PD, and the conclusions are controversial[17, 18].”, the authors need to discuss these prior studies and explain how they are controversial, and how they fit in the context of the present study.
- It would be better to mention the specific anti-PD drugs and their doses that the patients were taking. Also, did the authors exclude patients with co-existing infections or patients taking any other medications (anti-microbial, anti-lipemic, anti-anginal, anti-diabetic, anti-hypertensive etc.)? These can explain if there were any confounding effects of these drugs in the results of the study.
- The authors did not have any control subjects in their study. This makes it difficult to compare the effect seen in the subsets of PD patients with healthy controls. The authors need to explain why controls were not included in the study. This is particularly important because as mentioned in refs. 17 and 18, the earlier studies have found contrasting results when compared to the healthy controls. Also, no difference were seen among the subgroups of the PD patients; so it is very important to know if the PD group as a whole showed difference from the healthy population. In absence of control group, this remains unknown.
- The study did not evaluate the sex differences in the findings. It would be important to know if the male and female PD patients showed any difference or not. This will be important from clinical perspective as well.
- The statistical method section mentions “Continuous variables among the two groups (the control and OSA groups) were compared using..”. It is not clear what is meant by the control and OSA groups here. Also, the section should mention the significance of the “r” values (i.e. the degree of association/correlation: mild, moderate, severe). The values of “r” in most of the results indicate low degree of association, even when the “p” values were significant. The authors should explain the significance of this low degree of correlation, especially in Fig.1.
- The statement in section 4.1. “TD subgroup did have higher values for all autonomic parameters, indicating better function, and lower CASS, although the difference was not statistically significant” is very confusing. If there is no statistical difference between them, it is not clear what is meant by stating that a group has higher value than another.
- The study is inconclusive. The authors concluded that CAN severity did not correlate with the PD clinical scores or ESC parameters, and finally they suggested that future studies are required to see if autonomic functions differ among different subtypes of PD. Without having these conclusions sorted out, the study seems to be a very speculative one.
Author Response
- The abstract did not mention the conclusion and significance of the study.
Answer: The abstract has been modified, mainly in the last paragraph. In the conclusion of the abstract, the most important significance of the study was mentioned:
“Although the ESC was not strongly associated with the cardiovascular autonomic parameters, CAN risk score significantly correlated with parameters of cardiovascular autonomic function. The results suggest that CAN risk score may be used for screening of CAN in patients with PD before resorting to the more sophisticated and specific, but ultimately more time-consuming complete autonomic function testing.”
- With reference to the statement made in the introduction “There are few similar studies done in patients with PD, and the conclusions are controversial[17, 18].”, the authors need to discuss these prior studies and explain how they are controversial, and how they fit in the context of the present study.
Answer: We did discuss the discrepancy among our study and these two study in the Discussion section. (the last paragraph of P.15 to the first paragraph of P.16)
- It would be better to mention the specific anti-PD drugs and their doses that the patients were taking. Also, did the authors exclude patients with co-existing infections or patients taking any other medications (anti-microbial, anti-lipemic, anti-anginal, anti-diabetic, anti-hypertensive etc.)? These can explain if there were any confounding effects of these drugs in the results of the study.
Answer: The data of levodopa equivalent dose have been added to Tables 1, 3, and 4.
- The authors did not have any control subjects in their study. This makes it difficult to compare the effect seen in the subsets of PD patients with healthy controls. The authors need to explain why controls were not included in the study. This is particularly important because as mentioned in refs. 17 and 18, the earlier studies have found contrasting results when compared to the healthy controls. Also, no difference was seen among the subgroups of the PD patients; so it is very important to know if the PD group as a whole showed difference from the healthy population. In absence of control group, this remains unknown.
Answer: Thank you for your suggestion. We totally agree that adding a control group does enhance the value of the study and the difference between patients with Parkinson’s disease (PD) and normal subjects can be evaluated. However, the main aim of this study was to see if ESCs measured by Sudoscan can differentiate patients with CAN from patients without CAN in general PD population. Therefore, there was no control group in the original study design.
- The study did not evaluate the sex differences in the findings. It would be important to know if the male and female PD patients showed any difference or not. This will be important from clinical perspective as well.
Answer: There is no significant sex difference in ESC value according to previous study (ref. 32) and our own laboratory data (unpublished). We did not know if gender was used as a categorical variable in the algorithm of CAN risk score, but the analysis of the association between CAN risk score and autonomic function was not divided into subgroups of male and female in previous work by Yuan et. al (ref. 16). We had tried to do the statistical analyses dividing the patients into subgroups of male and female, and found that the results were similar. The finding suggested that sex difference may not be an important factor in this study background. As for the sex difference in cardiovascular autonomic function in patients with PD, the issue is beyond the scope of the current study.
- The statistical method section mentions “Continuous variables among the two groups (the control and OSA groups) were compared using..”. It is not clear what is meant by the control and OSA groups here. Also, the section should mention the significance of the “r” values (i.e. the degree of association/correlation: mild, moderate, severe). The values of “r” in most of the results indicate low degree of association, even when the “p” values were significant. The authors should explain the significance of this low degree of correlation, especially in Fig.1.
Answer: There were typos. We have corrected the statements in the sub-section in “statistical methods”. I am sorry for the typing mistakes. I the Result and Discussion section, we add the following sentences. They are as follows:
There was significant negative correlation between hand ESC and UPDRS subscore II and total scores (Figure 1A). As for the cardiovascular autonomic parameters, significant correlation only existed between VR and hand ESC (Figure 1B). Although, the correlation coefficient indicate a weak positive linear relationship (r<0.4).
Nevertheless, it is surprising that our data revealed UPDRS and parameters of cardiovagal function correlated with hand ESC instead of foot ESC although the correlation coefficient indicate a weak positive linear relationship (r<0.4).
- The statement in section 4.1. “TD subgroup did have higher values for all autonomic parameters, indicating better function, and lower CASS, although the difference was not statistically significant” is very confusing. If there is no statistical difference between them, it is not clear what is meant by stating that a group has higher value than another.
Answer: Thanks for your suggestion. We have modified the statements in a more tempered form: “Although the TD subgroup had lower CASS and higher values for all autonomic parameters, suggesting better autonomic function, the difference was not statistically significant.”
- The study is inconclusive. The authors concluded that CAN severity did not correlate with the PD clinical scores or ESC parameters, and finally they suggested that future studies are required to see if autonomic functions differ among different subtypes of PD. Without having these conclusions sorted out, the study seems to be a very speculative one.
Answer: The issues that you mentioned above do remain inconclusive according to our results. However, there was still some conclusions from the current study, as we described in the Conclusion section: “Although the ESC was not strongly associated with the cardiovascular autonomic parameters, CAN risk score significantly correlated with parameters of cardiovascular autonomic function. If a patient's CAN risk score value is higher than 33.5 (%), caution is needed with the presence of CAN even in PD patients who are asymptomatic. The results suggest that CAN risk score may be used for screening of CAN in patients with PD before resorting to the more sophisticated and specific, but ultimately more time-consuming complete autonomic function testing.”

Round 2
Reviewer 1 Report
The authors have provided sufficient revisions to address the Reviewers' concerns.
Author Response
Thanks for your comments.

Reviewer 2 Report
No additional comments
Author Response
Thanks for your comments.

Reviewer 3 Report
The authors have made substantial revisions that improved the quality of the paper. However, several issues remain.
- 1. Lines 39-40 and 311-312: “caution is needed with the presence of CAN even in PD patients who are asymptomatic” - What is meant by “caution is needed?” Please clarify this sentence.
- 2. Lines 78-79: “The successful translation of these approaches to clinics may offer the promise of improving the long-term outcome for patients with PD.” How do you propose that translating this device to the clinic would improve outcomes for PD patients? Please clarify.
- 3. The manuscript remains difficult to read due to use of passive voice. It would enhance the paper to extensively review and reduce passive voice in order to make sentences more concise and to clarify the findings.
- 4. Please review again for grammatical errors- some examples of these are listed below:
- Line 24-35: Incomplete sentence: “To examine the feasibity of electrochemical skin conductance (ESC) measurement as a screening service on CAN in patients with PD and investigate the severity of CAN among different motor phenotypes.” also, should be "feasibility" (spelling error)
- Line 93-9: "newly diagnosed with PD or were on follow-up for less than 6 months as their daily dose of anti-Parkinsonian agents was still under adjustment" - this is unclear, please clarify
- Line 201: should be “indicates”
- Line 231: should be “recommends”
- Line 245: should be “study by Xu et al.” (period instead of comma)
- Line 286: should be “consistent with”
- Tables 3 and 4: Age (year) should be (years)
Author Response
The authors have made substantial revisions that improved the quality of the paper. However, several issues remain.
- Lines 39-40 and 311-312: “caution is needed with the presence of CAN even in PD patients who are asymptomatic” - What is meant by “caution is needed?” Please clarify this sentence.
Answer: We have re-written the sentence as: “it is recommended to be aware of the presence of CAN even in PD patients who are asymptomatic.”
- Lines 78-79: “The successful translation of these approaches to clinics may offer the promise of improving the long-term outcome for patients with PD.” How do you propose that translating this device to the clinic would improve outcomes for PD patients? Please clarify.
Answer: We have re-written the sentence as: “The successful translation of these approaches to clinics may offer the awareness of associated CAN in patients with PD to improve the long-term outcome for these patients.”
- The manuscript remains difficult to read due to use of passive voice. It would enhance the paper to extensively review and reduce passive voice in order to make sentences more concise and to clarify the findings.
Answer: We try to do our best to make sentences more concise and to clarify the findings. A native English speaker also has revised the language of the manuscript.
- Please review again for grammatical errors- some examples of these are listed below:
Line 24-35: Incomplete sentence: “To examine the feasibity of electrochemical skin conductance (ESC) measurement as a screening service on CAN in patients with PD and investigate the severity of CAN among different motor phenotypes.” also, should be "feasibility" (spelling error)
Line 93-9: "newly diagnosed with PD or were on follow-up for less than 6 months as their daily dose of anti-Parkinsonian agents was still under adjustment" - this is unclear, please clarify
Line 201: should be “indicates”
Line 231: should be “recommends”
Line 245: should be “study by Xu et al.” (period instead of comma)
Line 286: should be “consistent with”
Tables 3 and 4: Age (year) should be (years)
Answer: We have checked the grammar again and all the above spelling or grammatical errors have been corrected. Thanks a lot.

Reviewer 4 Report
Thank you for addressing the reviewer's comments.
Author Response
Thanks for your comments.

Reviewer 5 Report
The authors have addressed my concerns.
Author Response
Thanks for your comments.
